# Virtual Water Flow and Water Footprint Assessment of an Arid Region: A Case Study of South Khorasan Province, Iran

**Ehsan Qasemipour [1]** and **Ali Abbasi [1,2,*]**

[1]   Department of Civil Engineering, Faculty of Engineering, Ferdowsi University of Mashhad, Mashhad 9177948974, Iran

[2]   Faculty of Civil Engineering and Geosciences, Water Resources Section, Delft University of Technology, Stevinweg 1, 2628 CN Delft, The Netherlands

*   Correspondence: aabbasi@um.ac.ir; Tel.: +31-15-2781029

**Abstract:** Water challenges—especially in developing countries—are set to be strained by population explosion, growing technology, climate change and a shift in consumption pattern toward more water-intensive products. In these situations, water transfer in virtual form can play an important role in alleviating the pressure exerted on the limited water resources—especially in arid and semi-arid regions. This study aims to quantify the 10-year average of virtual water trade and the water footprint within South Khorasan—the third largest province in Iran—for both crops and livestock products. The virtual water content of 37 crops and five livestock is first estimated and the water footprint of each county is consequently measured using a top-down approach. The sustainability of the current agricultural productions is then assessed using the water scarcity (WS) indicator. Results of the study show that in spite of the aridity of the study area, eight out of 11 counties are net virtual water exporters. Birjand—the most populous county—is a net virtual water importer. The 10-year average water footprint of the region is measured as 2.341 Gm$^3$ per year, which accounts for 2.28% of national water footprint. The region's average per capita water footprint however, with 3486 m$^3$, is 115% higher than the national ones. Crop production and livestock production are responsible for 82.16% and 17.84% of the total water footprint. The current intensive agricultural practices in such an arid region have resulted in a water scarcity of 206%—which is far beyond the sustainability criteria. This study gives the water authorities and decision-makers of the region a picture of how and where local water resources are used through the food trade network. The generated information can be applied by the regional policymakers to establish effective and applicable approaches to alleviate water scarcity, guarantee sustainable use of water supplies, and provide food security

**Keywords:** virtual water flows; water footprint; South Khorasan; agricultural products; sustainability

## 1. Introduction

Food production is inextricably linked to water resources [1] as more than 90% of the global water footprint is related to the agricultural production [2]. Uneven temporal and spatial distribution of water supplies—particularly in arid regions—represents a serious threat to their sustainability [3]. These areas can take advantage of trading large mass of food rather than producing their food requirements domestically to ameliorate their water availability. They import a large volume of water through importing foodstuffs virtually. Virtual water was first introduced by Allan [4] as the amount of water used in the production of a product. Many regions import or export a large volume of water virtually by collaborating in the national or international food trade system. Virtual water flows from regions with higher water productivity to areas with lower water endowments, which contributes to

water saving [1]. Developed countries are used to importing embodied water in goods from another countries [5]; in contrast, local authorities in developing countries—and Iran, in particular—encourage and subsidy farmers to expand agricultural practices, rather than limiting groundwater pumping, as an invaluable, natural infrastructure in facing natural shocks such as severe droughts. While such inducements in irrigated agriculture may trigger inefficient use of water supplies with high opportunity and environmental costs [6], this seems to be in favor of authorities to achieve complete self-sufficiency through exploiting from limited, and now seriously endangered groundwater tables [7].

Food trade network plays an essential role in achieving food security [8]. The environmental impacts of this network with regard to water [9,10], land [11,12], carbon emission [13,14] and socioeconomic impacts [15,16] confirm such importance. Since irrigated agriculture consumes about 40% of the global freshwater supplies [17], unsustainable use of blue water resources (i.e., surface and groundwater) will undoubtedly threaten the security of the food supply system [18]. Besides, climate change along with anthropogenic activities such as intensive agriculture endangers the availability of water resources in water-scarce regions and, consequently, will result in the situation of food insecurity [19]. Instead, food trade serves as a viable option of water saving [1,20] since it has been proved that the global trade in foodstuffs saved around 6% of the water used in agriculture [21].

Information associated with interregional virtual water trade network can be used to identify water-scarce regions with unequal trade patterns, through which they export their limited water resources mostly to water-abundant regions [22]. Many studies, accordingly, conducted to determine the direction of virtual water flows in an interregional scale [23–25]. The results will provide policymakers with sufficient information to reconfigure the trade pattern as an efficient tool in alleviating water scarcity. Virtual water concept, apart from its strategic application in solving water problems, has a number of other advantages [26]. As a result, it has been applied as a complementary tool not only in interregional water management [24,25,27,28], but also in global water assessment [1,10,13,29].

The aim of this study is to provide such valuable information considering the trade structure of food products, including crops and livestock in South Khorasan province, Iran. Dang et al. [30] pictured the virtual water transfer among the U.S. states, using network visualization software. In this study, however, the transfer data of originating and destination locations are not available, making this impossible. The food trade network of the region can illustrate the sustainability and vulnerabilities of local water resources. Understanding how water-food nexus empowers decision-makers to allocate the limited local water resources in such a way to reduce the vulnerability and, consequently increase the sustainability of water resources is important.

Regarding the main aim of the study, the virtual water content of 37 unprocessed crops and five primary livestock is quantified. Then a mass balance equation is used to estimate the virtual water flows of each county within the region. Finally, direct water consumption is taken into account to measure the water footprint of those counties.

## 2. Materials and Methods

### 2.1. Study Area

South Khorasan, the third biggest province by 95,385 square km, is located in the east of Iran and lies between 30°15′ N to 34°03′ N latitudes and 57°43 E to 61°04′ E longitudes. The province includes 11 counties: Birjand, Boshrooye, Darmian, Ferdows, Khusf, Nehbandan, Qaen, Sarayan, Sarbishe, Tabsa, and Zirkuh as shown in Figure 1. The climate is generally arid with a dry winter and hot summer by a temperature ranging from −13.57 °C in winter to 46.83 °C in summer [31]. Annual precipitation varies spatially with a minimum of 89 mm in Tabas to a maximum of 186 mm in Qaen. Most of the rainfall occurs from January to April. Due to the province's proximity to one of the hottest deserts in the world (Dasht-e-Lut), its relative humidity is low and limited to between 10% and 58%. Additionally, the evapotranspiration rate is considerably high due to the north-to-south wind speed of up to 6.5 m/s.

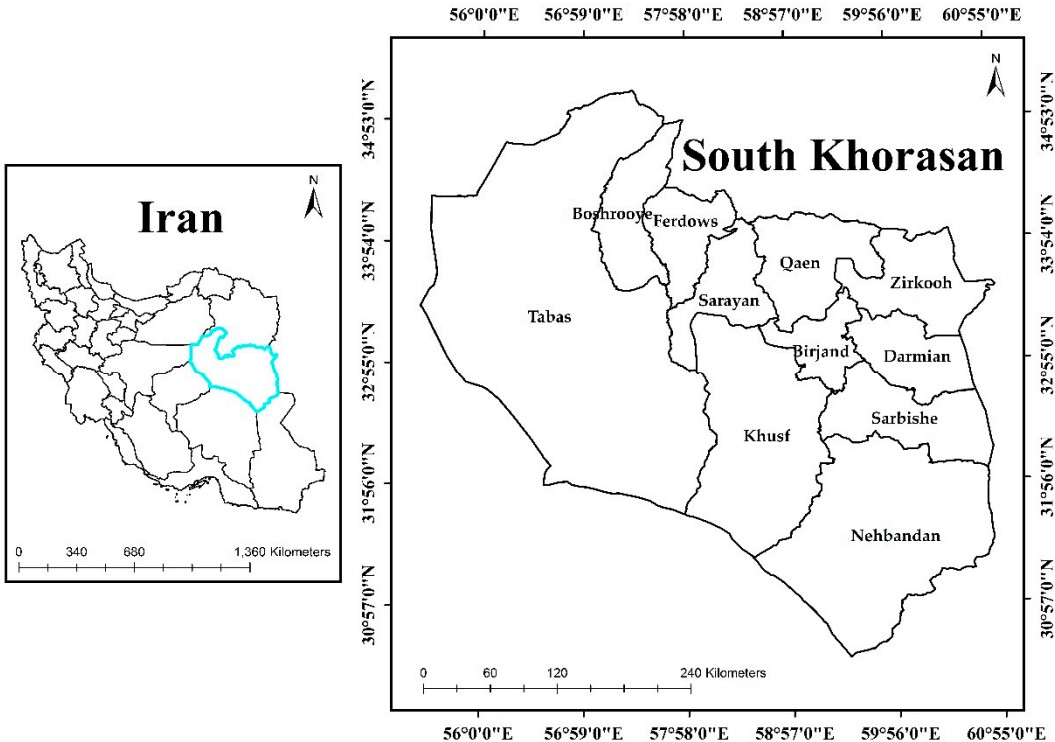

**Figure 1.** Map of the study area.

The spatial distribution of precipitation is shown in Figure 2 and confirms that most parts of the region have a little amount of annual rainfall (with an average of about 113 mm, less than 50% of the national average). Its temporal distribution, however, represents a more uneven distribution since most of the rainfall occurs in spring and the region experience no rain in almost six months of the year. This figure represents both spatial and temporal distribution of precipitation of South Khorasan based on the 10-year average data from 2006 to 2015.

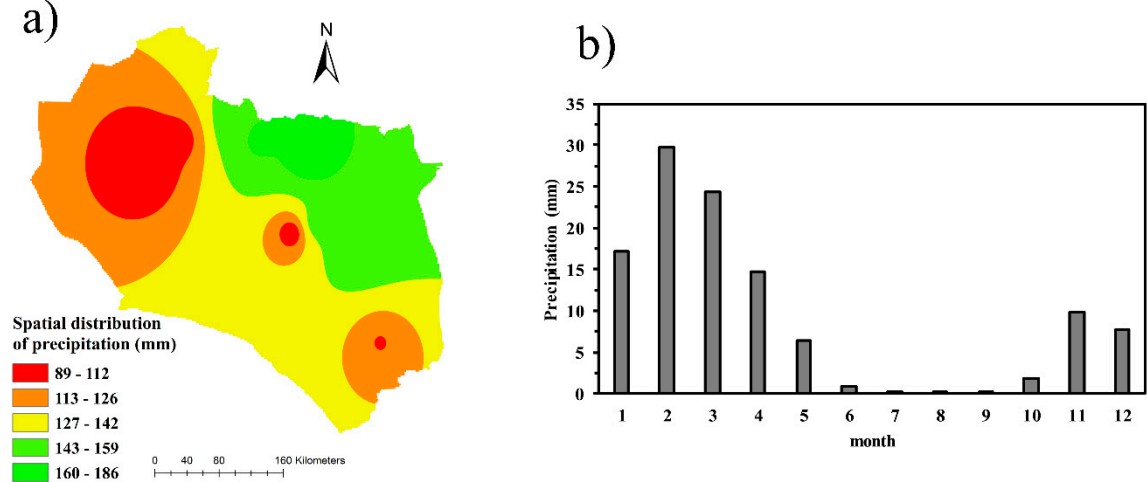

**Figure 2.** Spatial (**a**) and temporal (**b**) distribution of annual rainfall in South Khorasan.

### 2.2. Data Sources

Agricultural data, including planting dates, sowing area, crop production, crop yields, animal demographics, and livestock production is obtained from Agriculture Organization of the South Khorasan (AOSK). These data are available for 37 main crop types and five animal products for the

period of 2005 to 2014, as summarized in Table 1. According to the AOSK reports, these 37 crops account for nearly 80% of cultivated area in the province. These crops are classified into six groups of cereals, legumes, roots and fiber crops (henceforth only fiber crops is used), fruits, vegetables, and oilseeds. Transportation information of traded crops is not available, and as a result, the final destination of each crop type traded to and from the region is unknown. Therefore, tracing the amount of water virtually exported from or imported to the region is not possible. By employing the mass balance equation to see if the study area has a food surplus or deficit associated with its agricultural practices.

**Table 1.** Agricultural commodities included in this study.

| Commodity Group | Commodities | Short Name |
|---|---|---|
| Crops | Wheat, barley, maize, alfalfa, millet | Cereals |
| | Chickpea, mung bean, bean, lentil | Legumes |
| | Potato, sugar beet, turnip, cotton | Fiber crops |
| | Tomato, onion, cucumber, eggplant, zucchini, sweet melon, watermelon, garlic, cantaloupe | Vegetables |
| | Apple, pear, quince, sour cherry, cherry, plum, peach, apricot, table grape, pistachio, almond, walnut, carrot | Fruits |
| | Sesame, sunflower | Oilseeds |
| Livestock | The meat of sheep, goat, camel, and beef cows | Beef |
| | The meat of broiler hens | Chicken |
| | Egg of laying hens | Egg |
| | Honey | Honey |
| | Milk of sheep, goat, camel, and dairy cows | Milk |

CROPWAT is a decision support tool developed by the Food and Agriculture Organization of the United Nations (FAO), which is used to determine crop water requirements [32]. Using this model, Hoekstra and Hung [33] estimated the crop water requirements in different countries. In global assessments, climatic data of the capital or the most appropriate climatic station of each country is used [34]. Due to differences in climatic information within a country, those figures reported by them is not representative for the climate of other regions located in the country. Completing data regarding crop water requirements requires collecting data at a county-level, similar to what is done in this study.

*2.3. Methods*

In this section, the procedures of measuring the virtual water content of agricultural products, the virtual water flows among 11 counties within the province and the water footprint of each county are described. For this aim, we used 10-year average data of agricultural production during 2005–2014 to minimize some inconsistencies that were available in the obtained data.

2.3.1. Virtual Water Content of Crops

The virtual water content (VWC) of crops (m$^3$/ton) is determined as the ratio of the crop water requirement (m$^3$/ha) to the crop yield (ton/ha). Crop water requirement represents the volume of blue and green evapotranspiration, which can be determined by the popular CROPWAT model developed by the Food and Agriculture Organization of the United Nations (FAO). The required climatic data (e.g., monthly average of maximum and minimum temperature, relative humidity, monthly precipitation, wind speed, and sunshine hours) is obtained from South Khorasan Meteorological Organization [35]. In addition, the crop yield data is obtained from the Agriculture Organization of the South Khorasan province in a county-level. Following the procedures proposed by Hoekstra

and Hung [33], the crop water requirement of different crops in the region are calculated using the widely-used CROPWAT model:

$$ET_{blue} = \max(0, ET_c - P_{eff}), \tag{1}$$

$$ET_{green} = \min(ET_c, P_{eff}), \tag{2}$$

where $ET_{blue}$ and $ET_{green}$ represent the blue and green evapotranspiration (mm), respectively. Effective precipitation ($P_{eff}$) is a fraction of the total rainfall, which is available in the soil as the soil moisture and is used by the crop. Crop evapotranspiration ($ET_c$) refers to the crop water requirements under optimal conditions [4]. The VWC is then calculated as the sum of green and blue VWCs and is the ratio of crop water requirements (CWR, m$^3$/ha) to the crop yield (Y, ton/ha) as follows:

$$VWC_{blue} = \frac{CWR_{blue}}{Y}, \tag{3}$$

$$VWC_{green} = \frac{CWR_{green}}{Y}, \tag{4}$$

$$CWR_{blue} = 10 \times \sum_{n=1}^{lgp} ET_{blue}, \tag{5}$$

$$VWC_{green} = 10 \times \sum_{1}^{lgp} ET_{green}, \tag{6}$$

where lgp stands for the length of the growing period. The term 10 converts the depth of water (mm) into the volume of water per land surface (m$^3$/ha). The green ($CWR_{green}$) and blue ($CWR_{blue}$) crop water requirements represent the volume of water consumption from green (precipitation) and blue (surface and groundwater) water resources, respectively.

### 2.3.2. Virtual Water Content of Livestock Products

The procedures of calculating the virtual water content of livestock products are introduced by Chapagain and Hoekstra [36] as follows:

1. The virtual water content of live animals is quantified using Equation (7):

$$VWC_a = VWC_{direct} + VWC_{indirect}, \tag{7}$$

where $VWC_a$ is the virtual water content of the live animal $a$ (m$^3$/ton), $VWC_{direct}$ (or $VWC_{withdrawal}$) refers to the total volume of water needed for drinking, cleaning the environment, washing the animal, or other necessary services during the lifespan of the animal [36]. $VWC_{indirect}$ (or $VWC_{feed}$) is the volume of water used in producing the animal feed. Due to the lack of the data, it is assumed that the food needed for animals are provided within the province and the virtual water content of feed in the originating county is used to measure $VWC_{feed}$. As a result, $VWC_{indirect}$ for the same type of animals would be different in each county.

$VWC_{direct}$ accounts for a mere 1% of the total water footprint of livestock [30]. Following the procedures described by Mubako and Lant [37], direct and indirect virtual water content of live animals are estimated as follows:

$$VWC_{direct} = \frac{\int_{birth}^{slaughter} q_a \, dt}{W_a}, \tag{8}$$

$$VWC_{indirect} = \frac{\int_{birth}^{slaughter} \left\{ \sum_{c=1}^{n_c} VWC_c \times Feed_{a,c} \right\} dt}{W_a}, \tag{9}$$

where $q_a$, $VWC_c$, $Feed_{a,c}$, and $W_a$ are the daily volume of water consumption (m³/day), the virtual water content of feed c (m³/ton), the daily consumption of animal a of crop c (ton/day), and the average weight of live animal a before the slaughter (ton), respectively. The basic required information of the animal productions and the daily water consumptions are obtained from Agriculture Organization of South Khorasan [31].

2. The virtual water content of live animal ($VWC_a$) is then used to measure the virtual water content of animal products ($VWC_p$) such as beef, milk, egg, and chicken. In this study, only the products derived directly from a live animal (primary products) are considered. To calculate the virtual water content of livestock products the approach proposed by Chapagain and Hoekstra [36] is used in such a way that neither double-counting nor un-counting occurs.

$$VWC_p = (VWC_a + PWR_a) \times \frac{vf_p}{pf_p}, \tag{10}$$

where $PWR_a$ is the volume of water required for processing the primary products produced per ton of live animal a (m³/ton), $pf_p$ is the product fraction defined as the weight ratio of primary product p per tonnage of live animal, and $vf_p$ is the value fraction of product p defined as the ratio of the monetary value of product p to the sum of the market value of all products from that animal. The virtual water content of crops as well as those products directly derived from live animals (namely primary products) are presented in Table 2. In this study, five livestock products have been considered, including beef, chicken, egg, milk, and honey. In the case of chicken and egg, it should be noted that an average figure is used for estimating the indirect virtual water content since the number and amount of ingredients in the broiler and laying hens diet are negligible—as a result, the virtual water content of these products for all counties are the same as shown in Table 2.

**Table 2.** Virtual water content of agricultural products in counties (m³/ton).

| | Crops | | | | | | Livestock | | | | |
|---|---|---|---|---|---|---|---|---|---|---|---|
| **Regions** | **Cereals** | **Legumes** | **Fiber Crops** | **Vegetables** | **Fruits** | **Oilseeds** | **Beef** | **Chicken** | **Egg** | **Honey** | **Milk** |
| Birjand | 3841 | 10756 | 2752 | 1018 | 4320 | 20432 | 26956 | 4020 | 8213 | 0.15 | 5868 |
| Boshrooye | 2936 | 9032 | 2529 | 695 | 5644 | 11145 | 26057 | 4020 | 8213 | 0.15 | 5577 |
| Darmian | 3876 | 12587 | 1163 | 1004 | 5266 | 10446 | 29278 | 4020 | 8213 | 0.37 | 6621 |
| Ferdows | 2810 | 10826 | 2386 | 801 | 3715 | 10733 | 26395 | 4020 | 8213 | 0.19 | 5671 |
| Khusf | 3238 | 14083 | 3595 | 1527 | 5345 | 18096 | 29050 | 4020 | 8213 | 0.22 | 6507 |
| Nehbandan | 5093 | 20613 | 4061 | 863 | 5202 | 16099 | 34977 | 4020 | 8213 | 0.15 | 8443 |
| Qaen | 2611 | 6579 | 2456 | 910 | 3767 | 8976 | 24987 | 4020 | 8213 | 0.10 | 5236 |
| Sarayan | 3460 | 13446 | 3006 | 1020 | 5699 | 13719 | 29269 | 4020 | 8213 | 0.15 | 6591 |
| Sarbishe | 3516 | 14795 | 3403 | 891 | 5897 | 11861 | 29034 | 4020 | 8213 | 0.10 | 9505 |
| Tabas | 3139 | 6654 | 1789 | 447 | 4505 | 8559 | 25593 | 4020 | 8213 | 0.24 | 5438 |
| Zirkuh | 4005 | 12614 | 9375 | 1127 | 6216 | 14813 | 30614 | 4020 | 8213 | 0.11 | 7068 |

2.3.3. Virtual Water Transfer and Water Footprint Accounting

Virtual water flows of each county are calculated based on the amount of food transferred and using the virtual water concept. As the food transfer data in the region is not available for each county, the methodology described in Liu et al. [38] is applied to measure the food transfer amounts. In this approach, the agricultural products (i.e., crops and animal products) would be exported/imported if the amount of production is more/less than consumption in each county. For this aim, 10-year average data of both production and consumption are used to distinguish between exporting and importing counties. Then, interregional virtual water flows within the province is measured using Equation (11):

$$\text{VWF} = 10^{-6} \times \sum_{p=1}^{n_p} D_p \times \text{VWC}_p, \tag{11}$$

where VWF represents the annual virtual water flow (Mm$^3$), $D_p$ is the surplus or deficit of production (ton) of product p, and VWC$_p$ indicates virtual water content of the respective product (m$^3$/ton) in that county. Due to the lack of food transfer information regarding the selling and buying regions, it is assumed that in counties with food shortage, the water footprint of imported foods is equal to the virtual water content of that kind of product in the destination county. Following that, the water footprint of each county is quantified using the top-down approach proposed by Hoekstra and Hung [33] as follows:

$$\text{WF} = \text{WU} + \text{GVWI} - \text{GVWE}, \tag{12}$$

where WU, GVWI, and GVWE are the total domestic water use (direct water use) of a county, gross virtual water import and gross virtual water export (indirect water use), all in Mm$^3$ per year, respectively. Direct water consumption of inhabitance in each county is obtained from the Regional Water Company of South Khorasan province [39].

2.3.4. Sustainability Assessment of Agriculture

It would be a logical conclusion that a water-scarce region should rely on importing water-intensive and low value-added products rather than producing them domestically. On the other hand, however, exporting these commodities would provide an acceptable level of socioeconomic security in line with environmental sustainability for regions with an abundant supply of freshwater [10]. The water scarcity indicator (WS) is employed in this study to assess the agricultural sustainability of the region and is defined as the ratio of the withdrawn water to the available water (in a number of studies this indicator is called the Withdrawal-to-Availability (WTA) ratio). Water withdrawal here refers to the volume of water which is input in production while the volume of water removed from the producing region through evapo(transpiration), product integration, and discharge into other basins is called water consumption [40]. Some other newly devised indicators (WAVE + (Water accounting and Vulnerability Evaluation), AWARE (Available Water Remaining), and WSI (Water Stress Index)) might not result such comparable results. The reason is that these indices have bounded intervals of zero to one and due to the aridity of the study area, all counties shares the most unsustainable figure, one. The region can be classified as slightly (*WS* < 30%), moderate (30% ≤ *WS* < 60%), heavily (60% ≤ *WS* < 100%), and over (*WS* ≥ 100%) exploited [41].

**3. Results and Discussion**

*3.1. Virtual Water Content Estimates*

The virtual water content of six type of crops and five livestock products, as shown in Table 2, varies widely from county to county primarily due to different climatic conditions. Producing one ton of oilseeds in Birjand, for example, requires 127.6% more water than producing these types of crops in Qaen. Beef and oilseeds have the largest virtual water content (VWC) in agricultural

products. For crops, "vegetables" have the minimum VWC, which varies from 447 m$^3$/ton in Tabas to 1,527 m$^3$/ton in Khusf. Vegetables are responsible for only 5.62% of total water footprint, as shown in Figure 3. Among animal products, honey with a VWC of 0.18 m$^3$/ton on average has the highest water productivity, followed by chicken with 4,020 m$^3$/ton. Beef—as the most water-intensive animal product—is responsible for 21.57% of total water withdrawal, while it makes up 8.84% of total water footprint. Although cereals constitute 22.92% of the total water footprint of the region (Figure 3), they consume 28.84% of local water resources per year.

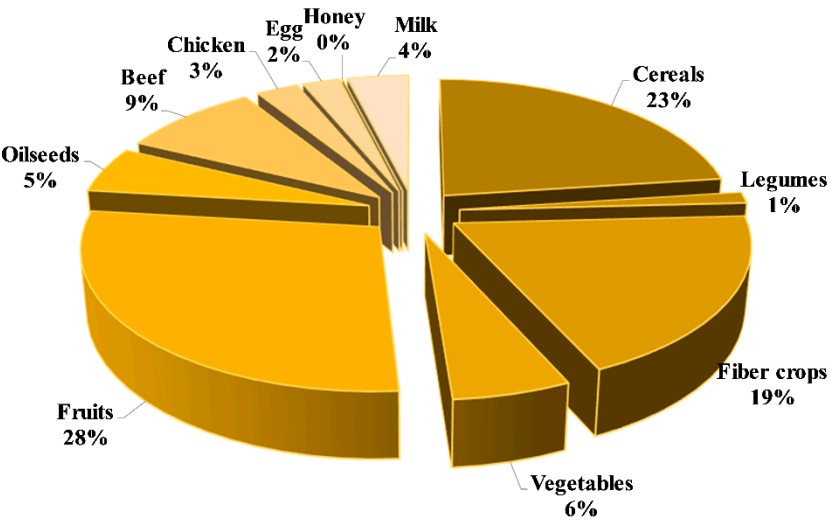

**Figure 3.** Contribution of each type of agricultural products in the water footprint of the province (total water footprint of the province is 2.34 Gm$^3$/year).

More than 95% of VWC figures reported in Table 2 are comprised of blue virtual water and, therefore, the study region relies heavily on its blue water resources to provide its food demands domestically—largely due to high rate of evapotranspiration and lack of sufficient rainfall. This has exerted an enormous amount of pressure on aquifers and contributed to overexploitation of groundwater resources, which in turn seriously threaten not only the environment, but also the socioeconomic situation of the region. Figure 3 shows the share of each agricultural products in the total water footprint of the region.

*3.2. Virtual Water Transfer*

The amount of virtual water transferred in the food products of the South Khorasan province is 2406 Mm$^3$. As shown in Figure 4, crops contribute to 96.40% of all imports, making the region a net virtual water importer. On the other hand, virtual water exports represented by animal products account for 77.43% of all exports, making the region a net virtual water exporter in terms of animal commodities. Fruits is the only type of crops that in all counties its production is much lower than the requirement. Therefore, it accounts for 35.63% of all virtual water imported (or 423 Mm$^3$). In livestock products, milk is responsible for the largest share of virtual water export with 528 Mm$^3$, accounting for 43.31% of all exports.

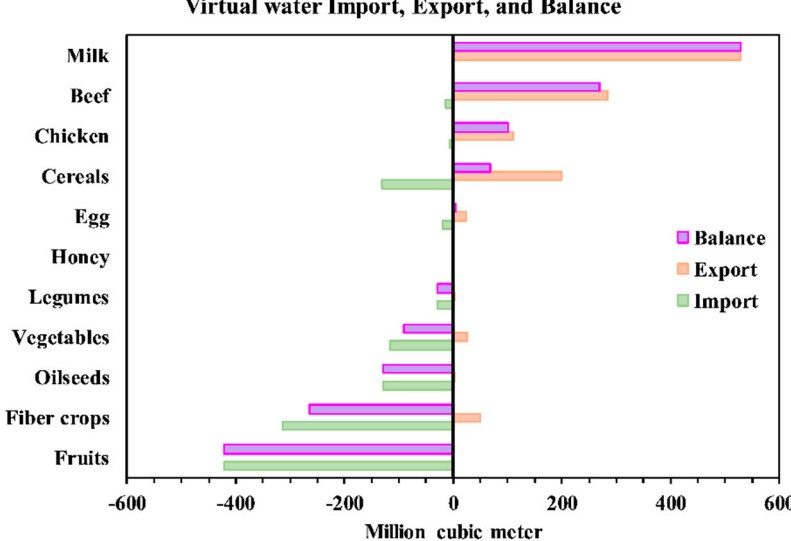

**Figure 4.** Virtual water import, export, and balance in agricultural products.

The virtual water transfer regarding the crops, livestock, overall, and per person are depicted in Figure 5a–d. Birjand is by far the major importer of virtual water by 551 Mm³ per year, followed by Zirkuh and Nehbandan by 121 Mm³ and 117 Mm³, respectively. In other side, Qaen, Boshrooye, Khusf, and Sarbishe are the major virtual water exporters with 235, 164, 149 and 132 Mm³ per year, respectively. Boshrooye and Sarayan are the only major virtual water exporters regarding crops. Birjand—as the largest virtual water importer regarding the crops—exports 111 Mm³ annually in animal products. In Table 3, the counties are ranked regarding their volume of virtual water imports/exports.

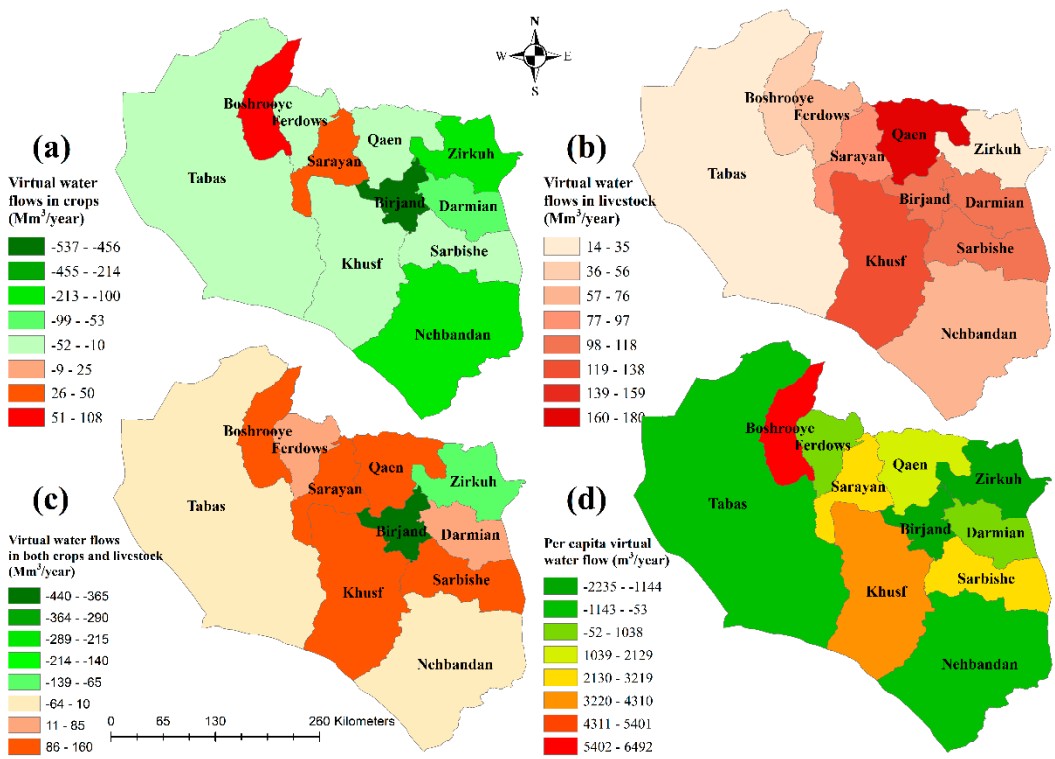

**Figure 5.** Virtual water flows associated with (**a**) crops; (**b**) livestock; (**c**) both of crops and livestock; and (**d**) per capita virtual water flows.

**Table 3.** Ranking the counties in virtual water import and export (Mm$^3$/year).

| Rank | Import | | Export | |
|:---:|:---:|:---:|:---:|:---:|
| 1 | Birjand | 551 | Qaen | 235 |
| 2 | Zirkuh | 121 | Boshrooye | 164 |
| 3 | Nehbandan | 117 | Khusf | 149 |
| 4 | Darmian | 94 | Sarbishe | 132 |
| 5 | Qaen | 83 | Sarayan | 115 |
| 6 | Tabas | 59 | Birjand | 111 |
| 7 | Ferdows | 55 | Darmian | 105 |
| 8 | Khusf | 47 | Nehbandan | 72 |
| 9 | Sarbishe | 44 | Ferdows | 72 |
| 10 | Sarayan | 12 | Tabas | 33 |
| 11 | Boshrooye | 3 | Zirkuh | 31 |

Figure 5d indicates per capita virtual water flow, which represents the virtual water flow intensity of each county. Boshrooye, for example, with 6492 m$^3$/year, has the largest per capita virtual water flow due to its intensive agriculture and small population. Birjand and Zirkuh, on the other hand, have the most significant volume of per capita virtual water imports.

Gross virtual water import and export of each type of crop and livestock products are presented in Tables 4 and 5 for all counties. Among crops, cereals is over-produced and a large amount of water (68 Mm$^3$) is virtually exported outside the province. In terms of crops, however, the region is a net virtual water importer with 869 Mm$^3$ per year. All counties participate in both interregional virtual water trade with a negative mark for imports and a positive mark for exports. Overall, the region is a net virtual water exporter with 1186 Mm$^3$ import and 1220 Mm$^3$ export per year. The imported virtual water includes 1144 Mm$^3$ for crops and around 43 Mm$^3$ for animal commodities. These amounts of the exported water are 275 and 944 Mm$^3$ for crops and animal commodities, respectively. From these numbers, 902 Mm$^3$ water is exported outside the local territory annually through exporting animal products. Instead, 869 Mm$^3$ water per year is imported from outside to meet crop demands of its inhabitance. Figure 6 illustrates the amount of water transferred in both crops and livestock products. It also shows the amount of both surplus and deficit in agricultural practices within the province and the difference between these two numbers is either imported from or exported to outside.

Due to the Iran's political situation and fighting against the sanctions, achieving absolute self-sufficiency in producing agricultural commodities, especially in strategic staple crops and cereals, is considered as a definite goal for local and national authorities. Although the region is a net virtual water importer in crops, current intensive agricultural practices exerted unprecedented pressure on groundwater resources, since the region lacks in green water resources [7].

**Table 4.** Comparison of 10-year average Gross Virtual Water Import (GVWI) and Gross Virtual Water Export (GVWE) of 11 counties in crops.

| Crops | | Birjand | Boshrooye | Darmian | Ferdows | Khusf | Nehbandan | Qaen | Sarayan | Sarbishe | Tabas | Zirkouh | Total |
|---|---|---|---|---|---|---|---|---|---|---|---|---|---|
| Cereals | $D_p$ (1000 ton) | −33.8 | 27.2 | −5.5 | 3.0 | 4.5 | 0.7 | 9.5 | 9.3 | 8.9 | 2.4 | −2.2 | 24.12 |
| | GVWI ($Mm^3$) | 101.0 | 0.0 | 21.2 | 0.0 | 0.0 | 0.0 | 0.0 | 0.0 | 0.0 | 0.0 | 8.9 | 131.10 |
| | GVWE ($Mm^3$) | 0.0 | 79.9 | 0.0 | 8.3 | 14.7 | 3.6 | 24.8 | 32.2 | 31.4 | 4.5 | 0.0 | 199.41 |
| | VWB ($Mm^3$) | −101.0 | 79.9 | −21.2 | 8.3 | 14.7 | 3.6 | 24.8 | 32.2 | 31.4 | 4.5 | −8.9 | 68.31 |
| Legumes | $D_p$ (1000 ton) | −1.0 | 0.0 | −0.2 | −0.2 | −0.1 | −0.2 | −0.4 | 0.0 | −0.2 | −0.3 | −0.2 | −2.70 |
| | GVWI ($Mm^3$) | 10.8 | 0.0 | 2.8 | 2.1 | 1.1 | 4.8 | 2.5 | 0.0 | 2.4 | 0.5 | 1.9 | 28.87 |
| | GVWE ($Mm^3$) | 0.0 | 0.3 | 0.0 | 0.0 | 0.0 | 0.0 | 0.0 | 0.3 | 0.0 | 0.0 | 0.0 | 0.03 |
| | VWB ($Mm^3$) | −10.8 | 0.3 | −2.8 | −2.1 | −1.1 | −4.8 | −2.5 | 0.3 | −2.4 | −0.5 | −1.9 | −28.84 |
| Fiber crops | $D_p$ (1000 ton) | −49.5 | 6.4 | −1.9 | −9.8 | −5.1 | −11.7 | 12.6 | 0.7 | −1.2 | −15.4 | −7.2 | −82.10 |
| | GVWI ($Mm^3$) | 136.2 | 0.0 | 2.2 | 23.4 | 18.5 | 47.6 | 0.0 | 0.0 | 4.0 | 15.5 | 67.2 | 314.58 |
| | GVWE ($Mm^3$) | 0.0 | 16.2 | 0.0 | 0.0 | 0.0 | 0.0 | 31.0 | 2.0 | 0.0 | 0.0 | 0.0 | 49.24 |
| | VWB ($Mm^3$) | −136.2 | 16.2 | −2.2 | −23.4 | −18.5 | −47.6 | 31.0 | 2.0 | −4.0 | −15.5 | −67.2 | −265.33 |
| Vegetables | $D_p$ (1000 ton) | −59.8 | 20.6 | −11.5 | −8.3 | −4.4 | −8.1 | −19.1 | −0.3 | −5.4 | −6.6 | 9.8 | −93.14 |
| | GVWI ($Mm^3$) | 60.9 | 0.0 | 11.5 | 6.6 | 6.8 | 7.0 | 17.4 | 0.3 | 4.8 | 0.9 | 0.0 | 116.26 |
| | GVWE ($Mm^3$) | 0.0 | 14.3 | 0.0 | 0.0 | 0.0 | 0.0 | 0.0 | 0.0 | 0.0 | 0.0 | 11.1 | 25.38 |
| | VWB ($Mm^3$) | −60.9 | 14.3 | −11.5 | −6.6 | −6.8 | −7.0 | −17.4 | −0.3 | −4.8 | −0.9 | 11.1 | −90.88 |
| Fruits | $D_p$ (1000 ton) | −37.9 | −0.5 | −8.5 | −3.7 | −2.6 | −8.2 | −13.3 | −0.9 | −4.4 | −10.9 | −4.6 | −95.65 |
| | GVWI ($Mm^3$) | 163.8 | 2.7 | 45.0 | 13.8 | 14.1 | 42.5 | 50.2 | 5.1 | 26.2 | 30.4 | 28.8 | 422.61 |
| | GVWE ($Mm^3$) | 0.0 | 0.0 | 0.0 | 0.0 | 0.0 | 0.0 | 0.0 | 0.0 | 0.0 | 0.0 | 0.0 | 0.00 |
| | VWB ($Mm^3$) | −163.8 | −2.7 | −45.0 | −13.8 | −14.1 | −42.5 | −50.2 | −5.1 | −26.2 | −30.4 | −28.8 | −422.61 |
| Oilseeds | $D_p$ (1000 ton) | −3.1 | 0.1 | −0.7 | −0.6 | −0.4 | −0.8 | −1.4 | −0.3 | −0.5 | −0.9 | −0.5 | −9.18 |
| | GVWI ($Mm^3$) | 64.2 | 0.0 | 7.6 | 6.4 | 7.0 | 12.2 | 12.8 | 4.0 | 6.2 | 1.9 | 7.9 | 130.12 |
| | GVWE ($Mm^3$) | 0.0 | 0.9 | 0.0 | 0.0 | 0.0 | 0.0 | 0.0 | 0.0 | 0.0 | 0.0 | 0.0 | 0.91 |
| | VWB ($Mm^3$) | −64.2 | 0.9 | −7.6 | −6.4 | −7.0 | −12.2 | −12.8 | −4.0 | −6.2 | −1.9 | −7.9 | −129.20 |

**Table 5.** Comparison of 10-year average Gross Virtual Water Import (GVWI) and Gross Virtual Water Export (GVWE) of 11 counties in livestock.

| Livestock | | Birjand | Boshrooye | Darmian | Ferdows | Khusf | Nehbandan | Qaen | Sarayan | Sarbishe | Tabas | Zirkouh | Total |
|---|---|---|---|---|---|---|---|---|---|---|---|---|---|
| Beef | $D_p$ (1000 ton) | −0.53 | 0.90 | 0.81 | 0.66 | 1.18 | 1.47 | 2.02 | 1.20 | 1.08 | 0.24 | 0.31 | 9.34 |
| | GVWI (Mm$^3$) | 14.41 | 0.00 | 0.00 | 0.00 | 0.00 | 0.00 | 0.00 | 0.00 | 0.00 | 0.00 | 0.00 | 14.41 |
| | GVWE (Mm$^3$) | 0.00 | 23.51 | 23.79 | 17.47 | 34.24 | 51.37 | 50.37 | 35.12 | 31.30 | 6.22 | 9.53 | 282.92 |
| | VWB (Mm$^3$) | −14.41 | 23.51 | 23.79 | 17.47 | 34.24 | 51.37 | 50.37 | 35.12 | 31.30 | 6.22 | 9.53 | 268.51 |
| Chicken | $D_p$ (1000 ton) | 7.22 | −0.11 | 6.39 | 1.05 | 3.98 | 0.53 | 3.78 | 0.30 | 3.92 | −1.16 | −0.66 | 25.24 |
| | GVWI (Mm$^3$) | 0.00 | 0.46 | 0.00 | 0.00 | 0.00 | 0.00 | 0.00 | 0.00 | 0.00 | 4.65 | 2.66 | 7.77 |
| | GVWE (Mm$^3$) | 29.04 | 0.00 | 25.71 | 4.24 | 15.99 | 2.12 | 15.18 | 1.20 | 15.77 | 0.00 | 0.00 | 109.24 |
| | VWB (Mm$^3$) | 29.04 | −0.46 | 25.71 | 4.24 | 15.99 | 2.12 | 15.18 | 1.20 | 15.77 | −4.65 | −2.66 | 101.47 |
| Egg | $D_p$ (1000 ton) | 0.68 | 0.04 | −0.51 | −0.31 | 0.10 | −0.33 | 0.32 | −0.32 | 1.78 | −0.64 | −0.40 | 0.43 |
| | GVWI (Mm$^3$) | 0.00 | 0.00 | 4.17 | 2.57 | 0.00 | 2.67 | 0.00 | 2.59 | 0.00 | 5.25 | 3.26 | 20.51 |
| | GVWE (Mm$^3$) | 5.57 | 0.30 | 0.00 | 0.00 | 0.85 | 0.00 | 2.66 | 0.00 | 14.66 | 0.00 | 0.00 | 24.05 |
| | VWB (Mm$^3$) | 5.57 | 0.30 | −4.17 | −2.57 | 0.85 | −2.67 | 2.66 | −2.59 | 14.66 | −5.25 | −3.26 | 3.55 |
| Honey | $D_p$ (1000 ton) | −0.13 | −0.02 | −0.04 | −0.03 | −0.01 | −0.04 | −0.07 | −0.02 | −0.03 | −0.04 | −0.03 | −0.44 |
| | GVWI (Mm$^3$) | 0.00 | 0.00 | 0.00 | 0.00 | 0.00 | 0.00 | 0.00 | 0.00 | 0.00 | 0.00 | 0.00 | 0.00 |
| | GVWE (Mm$^3$) | 0.00 | 0.00 | 0.00 | 0.00 | 0.00 | 0.00 | 0.00 | 0.00 | 0.00 | 0.00 | 0.00 | 0.00 |
| | VWB (Mm$^3$) | 0.00 | 0.00 | 0.00 | 0.00 | 0.00 | 0.00 | 0.00 | 0.00 | 0.00 | 0.00 | 0.00 | 0.00 |
| Milk | $D_p$ (1000 ton) | 13.09 | 5.10 | 8.34 | 7.35 | 12.86 | 1.81 | 21.26 | 6.63 | 6.01 | 4.09 | 1.51 | 88.04 |
| | GVWI (Mm$^3$) | 0.00 | 0.00 | 0.00 | 0.00 | 0.00 | 0.00 | 0.00 | 0.00 | 0.00 | 0.00 | 0.00 | 0.00 |
| | GVWE (Mm$^3$) | 76.82 | 28.47 | 55.20 | 41.66 | 83.68 | 15.28 | 111.31 | 43.70 | 39.10 | 22.22 | 10.64 | 528.08 |
| | VWB (Mm$^3$) | 76.82 | 28.47 | 55.20 | 41.66 | 83.68 | 15.28 | 111.31 | 43.70 | 39.10 | 22.22 | 10.64 | 528.08 |

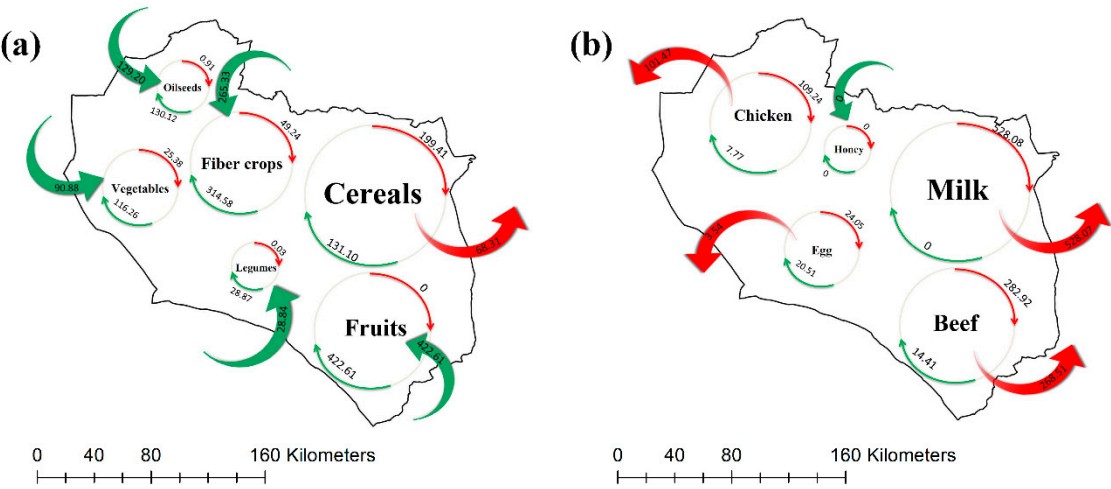

**Figure 6.** The inter and intraregional amount of virtual water transferred in crop (**a**) and livestock trade (**b**). Figures on the rotating arrows represent the internal virtual water flows, while the boundary arrows show the virtual water trade at the province level. Green and red colors refer to import and export, respectively. The size of circles represents the amount of water withdrawal in domestic production.

### 3.3. Water Footprint of Counties

Using the volumes of virtual water trade in agricultural productions as indirect water consumption and in combination with domestic water use as direct water consumption, the water footprint of each county can be measured (Equation (12)). From the sustainability point of view, all counties should be dependent on virtual water import, mainly due to their limited green (i.e., rainfall) and blue (i.e., irrigation) water resources [7]. The total water footprint and per capita water footprint of counties are presented in Figure 7. The most populous county—Birjand—has the greatest water footprint of about 728 Mm$^3$ per year, followed by Nehbandan and Zirkuh with 266 and 240 Mm$^3$ per year, respectively. Khusf, with only 71 Mm$^3$ per year, recorded the lowest water footprint within the region. Iran's water footprint is recorded as 102 Gm$^3$ per year and has an average per capita water footprint of 1624 m$^3$ [17]. In comparison, South Khorasan, as a whole, accounts for 2.28% of Iran's water footprint. The annual per capita water footprint within the study region (3486 m$^3$ per year) is nearly 114% more than the country, Iran (1624 m$^3$ per year). In addition, the annual per capita water footprint within the study region varies in a wide range from 1918 m$^3$ in Qaen to 5988 m$^3$ in Zirkuh. This high level of per capita water footprint is largely due to the aridity of the region, which substantially contributes to a higher virtual water content of agricultural products.

The region has an agriculture-based economy, and all the counties produce both crops and animal products extensively in comparison with their limited water endowments. Figure 8a compares the fractional contribution of different types of commodities in the agricultural productions. Intensive agricultural practices within the region have exerted enormous pressure on groundwater aquifers as the primary source of blue water resources in this arid region. The proportion of each source of water (i.e., green and blue), as well as the water availability of each county are presented in Figure 8b. According to the presented results, the region is overexploiting its blue water resources in crop production using withdrawal to availability (WTA) ratio. The readers are referred to Qasemipour and Abbasi [7] for more details on this issue.

Intensive agriculture, high virtual water contents of both crops and livestock, great reliance on blue water supplies, and insisting on complete self-sufficiency in producing foodstuffs has negatively affects the sustainability of the region. The water scarcity of the region is far beyond the scope of sustainability (Figure 9). The average water scarcity of the region is 206.06%, which means that the exploitation of groundwater aquifers is much higher than annual groundwater recharge.

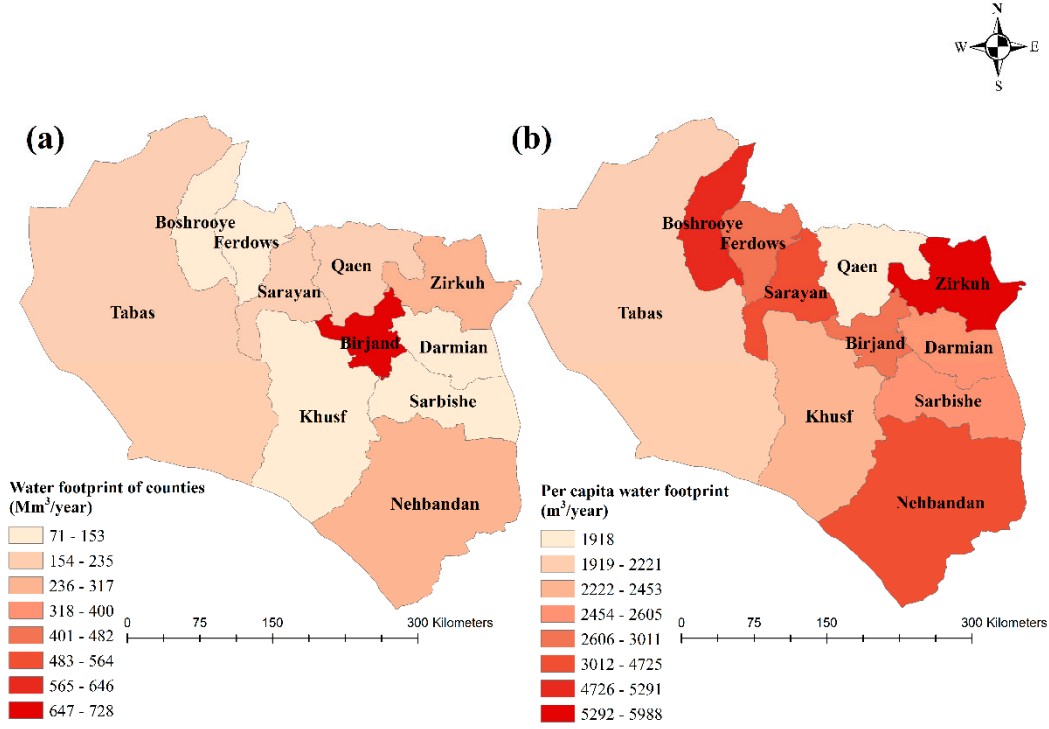

**Figure 7.** The water footprint of counties (**a**) and per capita water footprint (**b**).

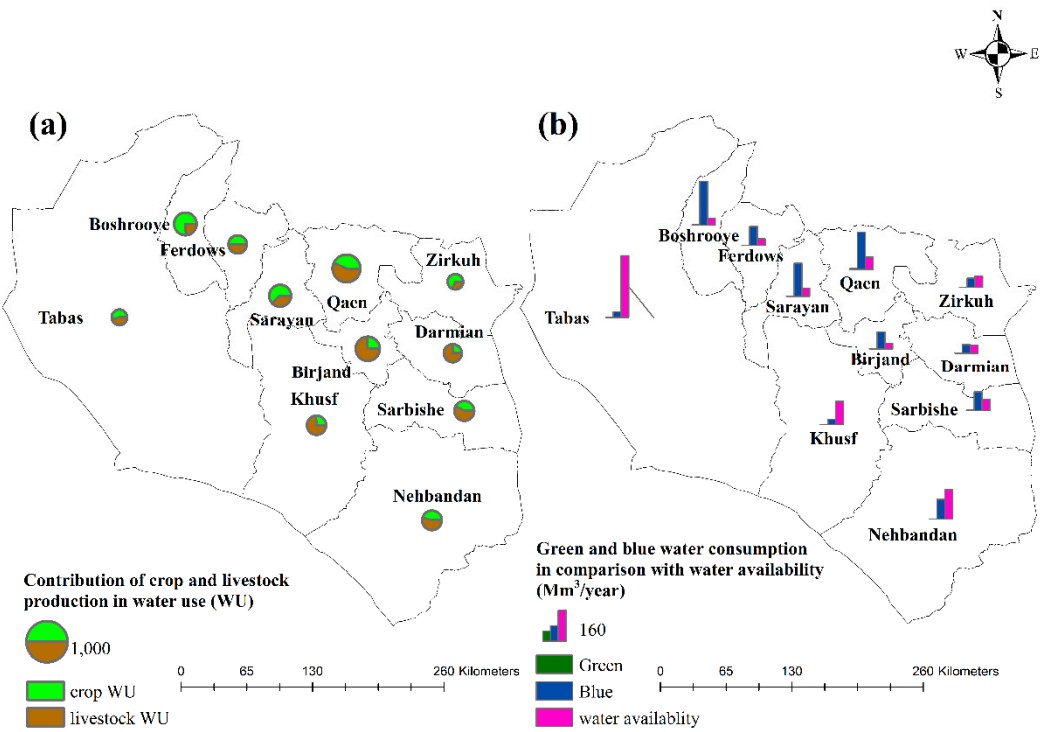

**Figure 8.** The fractional contribution of crop and livestock production in water withdrawal in each county (**a**), and green and blue water usage in comparison with water availability (**b**). The size of circles represents the total volume of water withdrawal.

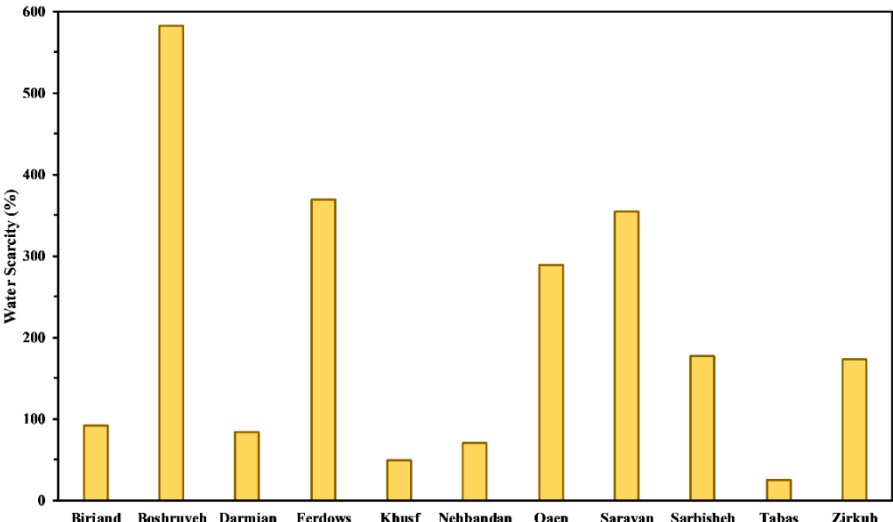

**Figure 9.** Water scarcity associated with agricultural production within the South Khorasan.

## 4. Conclusions

This study measures the virtual water transfer in agricultural produce and the water footprint of each county in the third biggest province in Iran, in which agriculture is the primary source of income. This is the first study quantifying interregional virtual water flows regarding both crops and livestock products in a region within Iran. Results show that South Khorasan is a net virtual water importer with regard to crops (Table 4), while it is a net virtual water exporter based on animal products (Table 5). As a whole, South Khorasan is a net virtual water exporter, which highlights the intensive livestock production within the region.

We suggest that the virtual water trade associated with livestock products is overestimated because a noticeable number of stocks in the region are not fed based on their balanced diet—which consists of ingredients with high virtual water content. On the other side, industrial and other services are not considered in this study, which means that the water footprint of the region is underestimated. Therefore, the estimates of virtual water transfer within the region face uncertainties inherent in both data acquired from respective organizations and water footprint accounting of agricultural products. It is highly suggested that further studies should be organized to meet these challenges.

The current trade system within the region is not sustainable (Figure 9) since such an arid region relies heavily on its groundwater resources to provide its inhabitance's food requirements and to make a profit from exports of food commodities as well. In such an arid region where irrigation is responsible for almost all the water used in agriculture, groundwater aquifers have a critical role to play in identity, culture and the economy of the region [42]. Due to the prolonged and severe droughts in the region, the VWC of crops and animal products are higher in comparison with figures for the country.

The structure of virtual water trade within the study area, as an arid and agriculture-based region, is also quantified. This information is important to understand the trade network and develop practical strategies in providing sustainable food security—particularly in the current situation of climate change and population growth. The findings of this study give a comprehensive picture of how trade structure is shaped within the region and helps policy-makers and water authorities to establish specific policies leading to water savings. They, for example, can improve water use efficiency through reconfiguring the trade system, adjusting the food market prices, and reallocating the water supplies to the most efficient and less water-consuming sectors in the economy of the region. Regional trade patterns of all regions in the country—along with the contribution of the country in the global food trade—await future studies as incomplete pieces of the puzzle of virtual water trade structure in Iran. Such studies are most likely to contribute to the local, national, and even international water and food security, purely in the sustainability point of view.

**Author Contributions:** Conceptualization, E.Q. and A.A; data curation, E.Q.; formal analysis, E.Q; methodology, E.Q. and A.A; project administration, A.A.; validation, A.A.; writing—original draft, E.Q.; writing—review and editing, A.A.

**Funding:** This research received no external funding.

**Conflicts of Interest:** The authors declare no conflict of interest.

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
