# Peer review of "Virtual Water Flow and Water Footprint Assessment of an Arid Region: A Case Study of South Khorasan Province, Iran"

_water, doi:10.3390/w11091755_

Round 1

Reviewer 1 Report

Manuscript water – 577509 “Virtual water flow and water footprint assessment of an arid region: a case study of South Khorasan Province, Iran”

Reviewer’s notes:

Although it is presented as a case study, your article is interesting in what concerns scientific soundness, originality, and methods. However, in my opinion, you should introduce some minor revisions, as follows:

1)     In Ln 37: “…90% of global water supplies is being used (consumed?) in food production”. Is this number a global (world) reference?  If yes, it should probably be lower, around 70 or 75%. Please verify.

2)     Ln 101: “This data is…” should be “These data are…”

3)     Ln 114-116: “Country” looks misused, instead of “county”

4)     Ln 134-135: “Blue” and “green” crop water requirements”, as components of ETc, are conventional, not universal, expressions, which should be explicitly defined and presented before common use, as “irrigation” and “effective precipitation” contributes to crop water requirements, respectively ETblue and ETgreen. 

5)     Ln 262: Revise the structure and contents of Table 3: export data are missing. Instead, counties identification is repeated.

6)     Ln 296, Tables 4 and 5: Dp(1000 ton) is not identified, and before it had not been, either.

7)     Although English is generally fine, a general edition can eliminate some point difficulties, getting the article quite clearer.

Reviewer 2 Report

The paper analyzes the virtual water flow and assesses the water footprint in South Khorasan Province, Eastern Iran. In my opinion, the submission is interesting and deserves to be published in the journal, especially if consider the fact that it addresses problems of water management in the agriculturally developed region with very limited water resources. However, there exist some shortcomings (mostly editorial), which require improvements or additional explanations before the final acceptance of the paper for publication. They are as follows:

1. Table 3: values of the virtual water export are missing. Moreover, please note that the 11 counties are arranged in exactly the same order for import and export, which means that they are of the same import and export ranks. I wonder if it is correct. Please confirm.

2. Figures 4, 5, 6, 7 and 8: the scale bar in kilometers should be added to the maps (similar to that in Figure 1).

3. Figure 6: it is suggested to put this figure in the “Study area” sub-chapter with relevant description of the spatial and temporal distribution of annual rainfall in the study area. Please add information about the multi-year period from which the data presented in the figure come from.

4. Please ensure high quality of the attached figures. For example, letters in Figures 2, 3 and 6 are too small and thus hardly readable. Please correct.

With the regard to the above-mentioned remarks it is recommended to accept the submission for publication after minor amendments.
